# Uncovering candidate genes responsive to salt stress in *Salix matsudana* (*Koidz*) by transcriptomic analysis

**Yanhong Chen[1]ᵒ, Yuna Jiang[1]ᵒ, Yu Chen[2]ᵒ, Wenxiang Feng[1], Guoyuan Liu[1], Chunmei Yu[1], Bolin Lian[1], Fei Zhong[1], Jian Zhang**[1]*

**1** Lab of Landscape Plant Genetics and Breeding, School of Life Science, Nantong University, Nantong, China, **2** College of Horticulture, Nanjing Agricultural University, Nanjing, China

ᵒ These authors contributed equally to this work.
* ntdxylzw@163.com

**Data Availability Statement:** All relevant data are within the manuscript and its Supporting Information files.

## Abstract

*Salix matsudana*, a member of *Salicaceae*, is an important ornamental tree in China. Because of its capability to tolerate high salt conditions, *S. matsudana* also plays an important ecological role when grown along Chinese coastal beaches, where the salinity content is high. Here, we aimed to elucidate the mechanism of higher salt tolerance in *S. matsudana* variety '9901' by identifying the associated genes through RNA sequencing and comparing differential gene expression between the *S. matsudana* salt-tolerant and salt-sensitive samples treated with 150 mM NaCl. Transcriptomic comparison of the roots of the two samples revealed 2174 and 3159 genes responsive to salt stress in salt-sensitive and salt-tolerant sample, respectively. Real-time polymerase chain reaction analysis of 9 of the responsive genes revealed a strong, positive correlation with RNA sequencing data. The genes were enriched in several pathways, including carbon metabolism pathway, plant-pathogen interaction pathway, and plant hormone signal transduction pathway. Differentially expressed genes (DEGs) encoding transcription factors associated with abiotic stress responses and salt stress response network were identified; their expression levels differed between the two samples in response to salt stress. Hub genes were also revealed by weighted gene co-expression network (WGCNA) analysis. For functional analysis of the DEG encoding sedoheptulose-1,7-bisphosphatase (SBPase), the gene was overexpressed in transgenic *Arabidopsis*, resulting in increased photosynthetic rates, sucrose and starch accumulation, and enhanced salt tolerance. Further functional characterization of other hub DEGs will reveal the molecular mechanism of salt tolerance in *S. matsudana* and allow the application of *S. matsudana* in coastal afforestation.

## Introduction

Soil salinity is one of the major adverse environmental factors that affects organ growth and productivity in plants. The land area affected by salt erosion has been increasing annually;

**Funding:** Jiang Zhang received the following three funds: National Natural Science Foundation of China (31971681, http://www.nsfc.gov.cn/), Jiangsu Province Forestry Science and Technology Innovation and Promotion Project (LYKJ [2018]36, http://jsf.jiangsu.gov.cn/) and Nantong University Scientific Research Start-up project for Introducing Talents (18R08, http://www.ntu.edu.cn/). The funders had no role in study design, data collection and analysis, decision to publish, or preparation of the manuscript.

**Competing interests:** The authors have declared that no competing interests exist.

**Abbreviations:** AP2/ERF, AP2-like ethylene-responsive transcription factor; DEGs, differentially expressed genes; FPKM, Fragments Per Kilobase of transcript per Million fragments mapped; NaCl, sodium chloride; qRT-PCR, Real-time Quantitative PCR; RNA-Seq, RNA-sequencing; ROS, reactive oxygen species; SBPase, Sedoheptulose-1,7-bisphosphatase; SnRK2, sucrose nonfermenting1-related protein kinase 2; SOS, salt overly sensitive; TFs, transcription factors; WT, wild-type.

approximately, one billion hectares of land are salt-affected, accounting for more than 6% of the world's total land area [1]. A high salt concentration leads to osmotic stress, accumulation of $Na^+$ and $Cl^-$ stress, and reactive oxygen species (ROS) production, all of which have negative effects on plant metabolism and growth, leading to a reduction in crop yield, with approximately US$ 27.3 billion losses every year [2, 3]. In China, it was estimated that approximately 30% of the salt-affected soil could be reclaimed by specific strategies to ensure food security and improve the economic environment [3, 4]. Salt-tolerant plant species selection and breeding would be the most efficient and direct strategy for reclamation of salt-affected soil. To obtain more salt-tolerant plants by genetic engineering, the first key step is to unravel the key components and molecular mechanisms of the salt-tolerance network in plants.

*Salix matsudana* Koidz, known as Chinese willow, is a member of the Salicaceae family and is native to northeastern China [5]. *S. matsudana* Koidz is an important ornamental and greening tree [6, 7]. The genome of most S. *matsudana* cultivars is heterotetraploid, which has higher salt tolerance than its diploid relatives. Hence, when grown along the Chinese coastal beach, where the salinity content is high, *S. matsudana* plays an important ecological role in improving the beach soil and alleviating the salinization. The newly reclaimed beach soil has higher salinity content and need new germplasm with higher salinity tolerance. Compared with other model plants and crops, only a few studies have illustrated the molecular mechanisms of stress tolerance in *Salix*. Using microarray analysis, a total of 403 salt-responsive genes was identified in *S. matsudana* by comparison of salt-treated roots and untreated controls [8]. The expression patterns of miRNAs and sHsp family of proteins and their potential roles in *Salix* salt tolerance were reported [9, 10]. Two genes, one coding for quinone reductase and the other for SpRLCK1, were identified as regulators of salt stress in *Salix*, and overexpression of the quinone reductase from *S. matsudana* Koidz enhanced salt tolerance in transgenic *Arabidopsis thaliana* [11, 12]. Further elucidation on the salt stress mechanisms in *Salix* is required to rapidly improve the salt-tolerant breeding and further application of the *Salix* sp.

With the development of second and third generation sequencing technology, RNA-seq has provided a low cost way to uncover molecular mechanisms in plant development and response to environmental signals by focusing on differential expression of genes and predicting key genes in the regulatory network [13–17]. Transcriptome sequencing or RNA-seq play an increasingly important role in the excavation of salt-tolerant genes in many plant species, such as cotton, rose, wheat and the woody plant *Jatropha curcas* [17–22]. For example, RNA-seq analysis of two citrus roots samples treated with salt stress for 4 and 24 h led to the identification of 454 overlapped differentially expressed genes (DEGs) [22]. Functional categorization of these DEGs revealed that some of them were involved in the salt overly sensitive (SOS) and ROS signaling pathways [22]. In addition, other DEGs coding for a variety of transcription factors (TF$_S$) have been identified, including WRKY, NAC, MYB, AP2/ERF, BZIP, which were verified as key regulators in salt or other stress signaling pathways in model plants and some crop plants [22].

Data obtained by RNA-seq have shed light on the salt stress tolerance regulatory network in many plants. However, to the best of our knowledge, a root transcriptome analysis on the *S. matsudana* varieties with different salt-tolerant capacities has not been performed. The sequencing of the complete genome of *S. matsudana* can facilitate the study of gene expression patterns. Hence, high-throughput RNA-seq provides an opportunity to study functional genomics of *S. matsudana*, and to uncover the specific molecular mechanisms underlying salt tolerance in this plant species.

Our previous study revealed two *S. matsudana* cultivars ('Yanjiang' and '9901') which exhibited different salt tolerance capabilities under salt stress [6], but it is not clear why these two cultivars have different tolerance traits. Therefore, this study aimed to understand the

molecular basis of genetic variation between salt-sensitive and salt-tolerant *S. matsudana* genotypes under salt stress through genome-wide transcriptome analysis, making this the first study to report the global transcriptome profile of *S. matsudana* roots.

## Materials and methods

### Phenotype analysis on *S. matsudana* 'Yanjiang' and '9901'

The stem cuttings (length 8–10 cm, coarse 2–3 mm) of 'Yanjiang' and '9901' samples were cultured in Hoagland solution with or without 100mM NaCl. The shooting and rooting time was recorded by observation every day during the 15 d of hydroponic cultivation. The shooting and rooting phenotypes were photographed using Nikon Z50 (Nikon, Tailand) and processed in Photoshop (Adobe).

### Plant materials and growth conditions

From the F1 generation seedlings of the cross between salt-sensitive cultivate *S. matsudana* 'Yanjiang' and salt-tolerant cultivate *S. matsudana* '9901', 12 salt-tolerant lines and 12 salt-sensitive lines were selected. In total, there were 15 lines, comprising the 12 F1 generation lines and 3 parent clones, in the salt-tolerant (ST) and salt-sensitive groups (SS), respectively. The 15 lines in each group was divided into 3 sub-groups (with one parent line included in each subgroup) and treated as three biological replicates.

The plant materials growth and treatment protocols were similar with the research previously reported [23]. The stem cuttings (length 8–10 cm, coarse 2–3 mm) of these samples were cultured for hydroponic rooting in Hoagland nutrient solution in greenhouse with a 16 h light/8 h dark cycle, day/night temperatures of 25˚C/ 20˚C, and relative humidity of 70%. After culture for 20 d, the roots were generated on the stem. The roots were treated with 150 mM NaCl dissolved in Hoagland solution; those not treated with NaCl were used as the control. The root samples were harvested at 4 h after salt stress treatment and immediately frozen in liquid nitrogen for RNA isolation and sequencing. The experiments on control and treatment samples in SS and ST group were repeated three times to avoid sampling errors. T01/ T02/T03 and T04/T05/T06 represent control (SS-CK) and treatment samples of salt-sensitive lines (SS-NT); T07/T08/T09 and T10/T11/T12 represent control (ST-CK) and treatment samples of salt-tolerant lines (ST-NT).

### RNA preparation, library preparation, and transcriptome sequencing

Using an RNeasy Plant Mini kit (Takara, Dalian, CN), total RNA from all samples was extracted following the manufacturer's instructions. DNA contamination was eliminated with DNase Ⅰ. RNA concentration was measured using NanoDrop 2000 (Thermo Scientific, Waltham, USA). RNA integrity was assessed using the RNA Nano 6000 Assay Kit of the Agilent Bioanalyzer 2100 system (Agilent Technologies, CA, USA).

A total of 1 μg RNA per sample was used as input material for the RNA sample preparations. Sequencing libraries were generated using NEB Next Ultra™ RNA Library Prep Kit for Illumina (NEB, USA) following the manufacturer's recommendations and index codes were added to attribute sequences to each sample. Briefly, it involved a series of procedures, including mRNA purification, first and second cDNA strand synthesis, adaptor ligation, PCR amplification, and cluster generation. The library preparations were sequenced on an Illumina Hiseq Xten platform and paired-end reads were generated. Clean data (clean reads) were obtained by removing reads containing adapter, reads containing ploy-N, and low quality reads from the raw data. At the same time, Q20, Q30, GC-content, and sequence duplication

level of the clean data were calculated. All the downstream analyses were based on clean data with high quality.

## Comparative analysis and gene functional annotation

The clean reads were mapped to the reference genome sequence of *S. matsudana* using TopHat2 software [24]. Only reads with a perfect match or one mismatch were further analyzed and annotated based on the reference genome. Gene function annotation was performed by sequence similarity searches using the BLAST program against the following databases: Nr (NCBI non-redundant protein sequences); Nt (NCBI non-redundant nucleotide sequences); Pfam (Protein family); KOG/COG (Clusters of Orthologous Groups of proteins); Swiss-Prot (A manually annotated and reviewed protein sequence database); KO (KEGG Ortholog database); and GO (Gene Ontology) [25].

## Gene expression and DEGs identification

Gene expression levels were estimated by fragments per kilobase of transcript per million fragments mapped (FPKM). Differential expression analysis of two conditions/groups was performed using the DESeq R package (1.10.1). To identify DEGs, fold change $\geq 2$ and false discovery rate (FDR) < 0.01 were used as screening criteria. The multiple of difference (fold change) was the ratio of the expression amount between two groups, and the FDR was obtained by correcting the p-value of difference significance. Because the differential expression analysis of transcriptome sequencing is an independent statistical hypothesis test for a large number of gene expression values, there will be false-positives. Therefore, the Benjamin Hochberg correction method was used to correct the significance p-value obtained from the original hypothesis test, and the finally FDR was used to identify the DEGs [26].

## GO and KEGG pathway analysis of DEGs

The DEGs were mapped into GO database for GO analysis. The GO annotations of sequences were extracted using Blast2GO [27, 28]. The KOBAS [29] software was used to test the statistical enrichment of DEGs in KEGG pathways [29].

## Validation of DEGs by qRT-PCR

To validate the expression pattern of DEGs, the candidate unigenes were selected for real-time RT-PCR analysis. The RNA samples used for sequencing were also used as template for qRT-PCR and the analysis was carried out as reported previously [30]. The specific primers were designed with primer 6 software and the ubiquitin gene was used as an internal control. The primer sequences are listed in S1 Table.

## Co-expression gene network construction on DEGs

Weighted gene co-expression network (WGCNA) analysis was performed as described previously and visualized by Cytoscape software [31, 32].

## Expression pattern analysis of genes related to salt response pathway

The *S. matsudana* homolog genes in the *Arabidopsis* salt sensory pathway, including genes of $Ca^{2+}$-signaling pathway proteins, high-affinity $K^+$ transporter (HKT), $Na^+/H^+$ exchangers (NHX), and the SOS $Na^+$ transporter were found from the expression data and their expression patterns were represented using heat map. The interaction network of *Arabidopsis* homologs of these genes was analyzed using Arabidopsis interactions views (http://bar.utoronto.ca/

interactions/cgi-bin/arabidopsis_interactions_viewer.cgi) and illustrated by Cytoscape software [32]. Hidden Markov Model (HMM) files of five TFs (NAM, PF02365; MYB, PF00249; AP2/ERF, PF00847; bZIP, PF00170; WRKY, PF03106) were acquired from the plant transcription factor database (http://pfam.xfam.org/) and the hmmsearch was carried out by searching the *S. matsudana* protein database using an in-house Perl script to find the members of the five TF families. The expression patterns of the putative DEGs of the TFs in the RNA-seq were presented by heatmaps using TBtools [33].

## Functional analysis of candidate gene using transgenic *Arabidopsis*

The CDS sequence of the candidate gene SBPase was cloned in a plant expression vector under the control of 35S enhancer, and the vector was transformed into *Arabidopsis* (*Col-0* ecotype) via *Agrobacterium*-mediated transformation protocol. The transgenic plants were screened and verified as previously reported [30]. Two independent lines of transgenic plants (L20, L22) were selected for this study. The plant growth condition and salt stress experiments were set up as previously reported [34]. The SBPase enzyme activity, total soluble sugar, sucrose, and starch contents were measured using kits according to the manufacturer's instructions (Solarbio, Beijing, CN). The relative electrolyte leakage was determined by the manual reported by Cao [35]. The seeds of WT and two transgenic Arabidopsis lines were planted on MS medium supplemented with 0mM NaCl, 50mM NaCl, 75mM NaCl, 100mM NaCl respectively and cultured for 7 days, the phenotype differences were recorded.

## Results

### Difference phenotypes in salt tolerance between the *S. matsudana* '9901' and 'Yanjing' varieties

Stems of the two *S. matsudana* '9901' and 'Yanjing' varieties were cultured in hydroponic solution containing 100 mM NaCl for 15 days. The shoot development was recorded every day. The stems of '9901' began to shoot on the third day. The shoot number and the shoot length in '9901' were prominently higher and longer than those in the 'Yanjiang' variety, which suggested that '9901' line possessed a stronger salt resistance than 'Yanjiang' (Fig 1).

### Sequencing and quality control

To gain comprehensive insights into the *S. matsudana* transcriptomic response to salinity stress, a total of 12 libraries, comprising salt-sensitive and salt-tolerant groups grown under different salinity treatment conditions, were constructed in this study. After cleaning and quality control, a total of 83.81 Gb sequencing data, with more than 283,192,303 paired-end reads, were obtained by Illumina Hi-seq 2000 platform (Sequence Accession Number). The Q30 value (sequencing error of 0.1%) was no less than 89.21%, and the GC percentage was between 44% and 45.5% in all 12 samples. Using the whole genome sequence of *S. matsudana* as the reference genome, the alignment results showed that 72.49%–74.47% of the total reads mapped to the reference genome, of which, approximately 70% reads mapped to the exon region and 19% mapped to the intergenic region (Table 1). These results were similar to those obtained in a previous study [36], indicating a good quality of the transcriptome.

### Identification and annotation of novel candidate genes

Based on the selected reference genome sequence, the mapped reads were assembled using the Cufflinks software, and compared with the original genome annotation information to find the original uncommented transcription area, and explore the new transcripts and genes of the

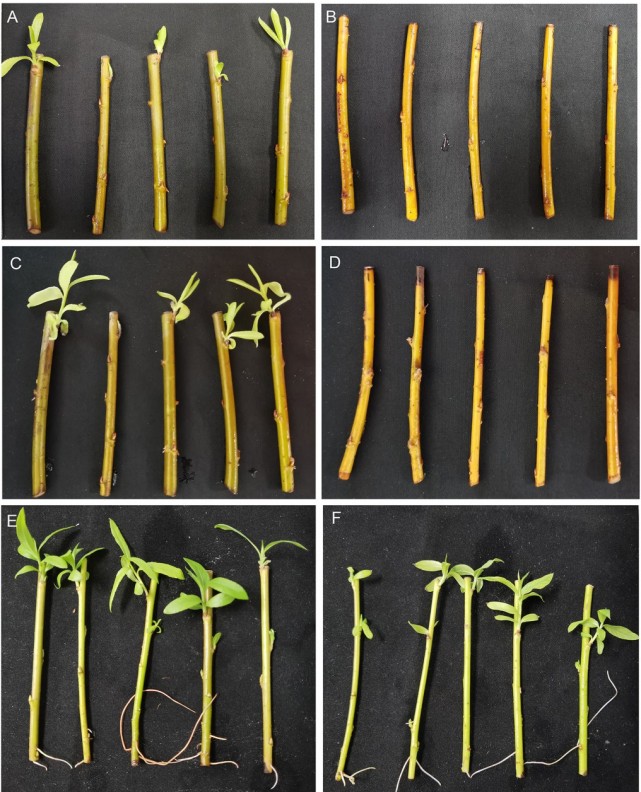

**Fig 1. '9901'material possessed stronger salt resistance than 'Yanjiang'.** Stems of '9901' and 'Yanjiang' varieties were cultured via hydroponics in Hoagland medium supplemented with or without 100 mM NaCl. The time for shoot development and number of shoots were assessed every day, and the phenotype was recorded on day 9 and 15. (A), '9901', 9 d under salt treatment; (B), 'Yanjiang', 9 d under salt treatment; (C), '9901',15 d under salt treatment; (D), 'Yanjiang', 15 d under salt treatment. (E), '9901', 15 d under normal condition; (F), 'Yanjiang', 15 d under normal condition.

**Table 1. Evaluation and genome-wide comparison of sequencing data.**

| Samples | Read Number | Base Number | GC Content | %≥Q30 | Total Reads | Mapped Reads | Mapped Ratio |
|---------|-------------|-------------|------------|-------|-------------|--------------|--------------|
| T01 | 25,578,589 | 7,531,508,862 | 44.85% | 90.13% | 51,157,178 | 37,866,865 | 74.02% |
| T02 | 26,421,335 | 7,797,364,710 | 45.15% | 90.62% | 52,842,670 | 38,735,898 | 73.30% |
| T03 | 22,245,921 | 6,579,520,468 | 45.32% | 90.04% | 44,491,842 | 32,253,487 | 72.49% |
| T04 | 22,257,488 | 6,577,183,488 | 44.56% | 89.95% | 44,514,976 | 32,791,358 | 73.66% |
| T05 | 21,538,540 | 6,396,186,834 | 44.97% | 89.43% | 43,077,080 | 31,608,285 | 73.38% |
| T06 | 21,252,245 | 6,315,759,964 | 45.37% | 89.34% | 42,504,490 | 31,619,362 | 74.39% |
| T07 | 25,104,772 | 7,454,658,666 | 45.00% | 90.26% | 50,209,544 | 37,060,546 | 73.81% |
| T08 | 25,141,779 | 7,487,006,450 | 45.45% | 89.71% | 50,283,558 | 36,796,323 | 73.18% |
| T09 | 23,802,232 | 7,064,283,298 | 44.98% | 90.26% | 47,604,464 | 35,294,748 | 74.14% |
| T10 | 23,095,949 | 6,857,514,622 | 45.20% | 89.44% | 46,191,898 | 33,981,128 | 73.57% |
| T11 | 21,350,772 | 6,334,276,966 | 45.26% | 89.21% | 42,701,544 | 30,989,791 | 72.57% |
| T12 | 25,402,681 | 7,418,748,980 | 45.00% | 90.89% | 50,805,362 | 37,836,188 | 74.47% |

species, so as to supplement and improve the original genome annotation information. A total of 5568 new genes were found by filtering out the sequences of short peptide chains (less than 50 amino acid residues) or single exons. The BLAST software was used to compare the new genes with NR, Swiss-Prot, GO, COG, KOG, Pfam, and KEGG databases, kobas2.0 was used to carry out KEGG ontology analysis, and the HMMER software was used to compare with Pfam database after predicting the amino acid sequences of the new genes. Finally, 5,181 new genes were functionally annotated.

## Identification of DEGs under high salinity

Gene expression levels were determined by calculating the number of clean reads mapped to the reference database for each gene (read count) and then normalizing to the FPKM value. The FPKM values of all genes are listed in S2 Table. Fold change ≥ 2 and FDR < 0.01 were used as screening criteria to select DEGs in three comparison combinations, namely SS-CK vs. SS-NT; ST-CK vs. ST-NT; SS-NT vs. ST-NT.

A total of 2174 DEGs were identified to be significant in SS-CK vs. SS-NT comparison, and these comprised 1787 upregulated and 387 downregulated genes. Furthermore, 3159 significant DEGs were detected in ST-CK vs. ST-NT comparison, with 2342 genes upregulated and 817 genes downregulated. Only 9 DEGs were found when SS-NT was compared with ST-NT, and these comprised 8 upregulated and 1 downregulated gene (Table 2).

To represent the DEGs of the three pair-wise comparisons, we created three heatmaps of FPKM-normalized transcript isoforms [log10 (FPKM+0.000001)] through hierarchical clustering (S1 Fig). The unique and shared DEGs in each group are shown using Venn diagrams (Fig 2). Two comparison combinations, SS-CK vs. SS-NT and ST-CK vs. ST-NT shared 1489 DEGs. Only 1 and 2 DEGs were found to be shared in the other two comparison combinations respectively. However, no DEGs were found to be shared among the three comparison combinations.

## GO and KEGG enrichment analysis of salt-responsive DEGs

A total of 2094 DEGs from SS-CK vs. SS-NT combination could be annotated in six databases, including 1411 genes (67.4%) annotated in the GO library. There were 3042 DEGs from ST-CK vs. ST-NT combination that showed annotated information in the databases, including 2094 genes (68.8%) annotated in the GO database. From 9 DEGs found in SS-NT vs. ST-NT, 7 DEGs was annotated in the GO library (Table 3).

To gain insights into the functional categorization and metabolic pathways involved in salt tolerance of *S. matsudana*, the DEGs identified in this study were subjected to enrichment analysis based on GO and KEGG databases. The results of enrichment analysis by GO library indicated that the genes were involved in all three major functions: the biological process,

**Table 2. Number of identified DEGs in three comparisons.**

| DEG_Set | All_DEG | up-regulated | down-regulated |
|---|---|---|---|
| SS-CK vs.SS-NT | 2174 | 1787 | 387 |
| SS-NT vs. ST-NT | 9 | 8 | 1 |
| ST-CK vs. ST-NT | 3159 | 2342 | 817 |

SS-CK and SS-NT indicate the salt-sensitive group samples treated without and with 150 mM NaCl; ST-CK and ST-NT indicate the salt-tolerant group samples treated without and with 150 mM NaCl. DEGs, differentially expressed genes.

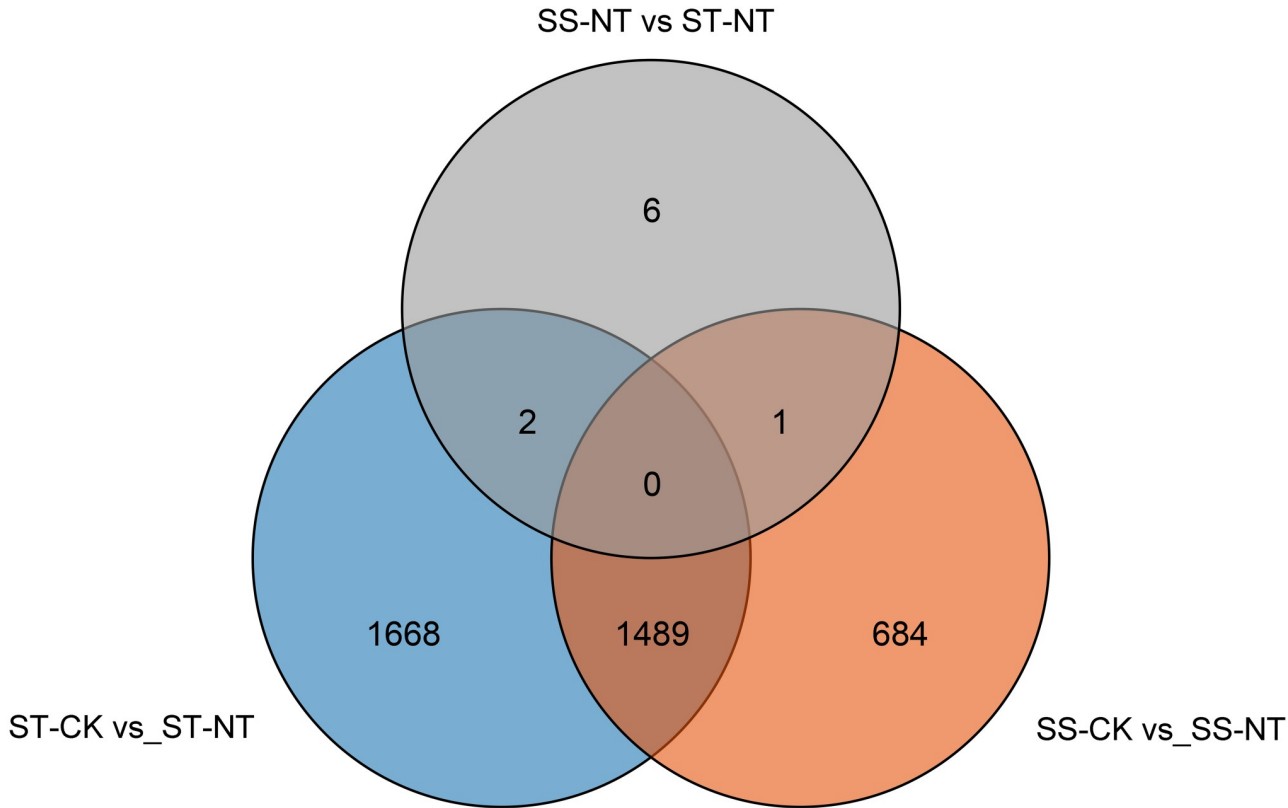

**Fig 2. Venn diagram shows the number of DEGs and overlapping DEGs in three comparison combinations.** SS-CK and SS-NT indicate the salt-sensitive group samples treated without and with 150 mM NaCl; ST-CK and ST-NT indicate the salt-tolerant group samples treated without and with 150 mM NaCl.

cellular component, and molecular function. Further data analysis on the GO secondary node annotation suggested that all DEGs could be divided into 53 functional groups. In the classification of biological processes, 'metabolic process' was the most significantly enriched term, followed by 'cellular process'; in the cell group classification, 'cell difference' was the most enriched, followed by 'cell'. In molecular function, most terms were related to 'binding', followed by 'catalytic activity' (Fig 3).

The DEGs were further mapped to the KEGG database and their enrichment in metabolic pathways and signaling pathways were analyzed to obtain more information on the difference in DEGs between salt-sensitive samples and salt-tolerant samples under salinity stress. The annotation results could be classified into five categories, namely cellular processes, environmental information processing, genetic information processing, metabolism, and

**Table 3. Statistics of the number of annotated DEGs.**

| DEG_Set | Annotated | COG | GO | KEGG | Swiss-Prot | eggNOG | NR |
|---|---|---|---|---|---|---|---|
| SS-CK vs. SS-NT | 2094 | 710 | 1411 | 637 | 1607 | 205 | 2093 |
| SS-NT vs. ST-NT | 9 | 4 | 7 | 8 | 8 | 3 | 9 |
| ST-CK vs. ST-NT | 3042 | 1045 | 2094 | 982 | 2274 | 322 | 3039 |

The footnote is same as Table 2.

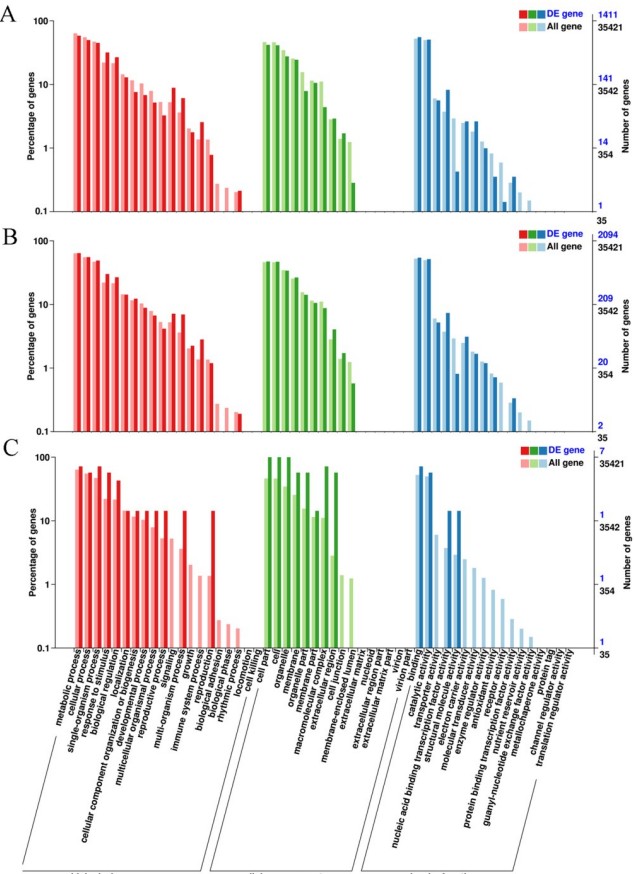

**Fig 3. GO annotation classification of the annotated DEGs.** This figure shows the enrichment of each DEG with respect to the secondary function in GO classification, the left side of the ordinate is the percentage of gene number, and the right side is the gene number. A, GO annotation classification of DEGs from the comparison SS-CK vs. SS-NT; B, GO annotation classification of DEGs from the comparison ST-CK vs. ST-NT; C, GO annotation classification of DEGs from the comparison SS-NT vs. ST-NT. DEGs, differentially expressed genes; GO, Gene ontology; SS-CK and SS-NT indicate the salt-sensitive group samples treated without and with 150 mM NaCl; ST-CK and ST-NT indicate the salt-tolerant group samples treated without and with 150 mM NaCl.

organizational systems, of which the metabolism category has the largest number of subcategories. A total of 58 DEGs (15%) from salt-sensitive samples were mapped to the top enriched term of 'plant hormone signal transduction' under the environmental information response pathways. The second enriched pathway was the plant-pathogen interaction from the biological system pathways, with 42 DEGs (11.5%) annotated in this pathway (Fig 4).

Of the 2094 DEGs identified from the salt-tolerant samples, 57 DEGs (9.5%) were classified in the first rank enrichment pathway, plant-pathogen interaction pathway. The second enrichment pathway was plant hormone signal transduction, with 53 DEGs (9.5%) found in this pathway. The third enrichment pathway was starch and sucrose metabolism, which was also found in the annotation of DEGs from salt-sensitive samples (Fig 4).

Nine DEGs were identified in pair-wise combination SS-NT vs. ST-NT, their differential expression pattern is illustrated in S1C Fig and Fig 5. The annotation of GO, KEGG pathway and NR of these DEGs are listed in S3 Table. Five genes were classified in the metabolism pathway, and four genes were found in the carbon metabolism pathway. In the NR annotation,

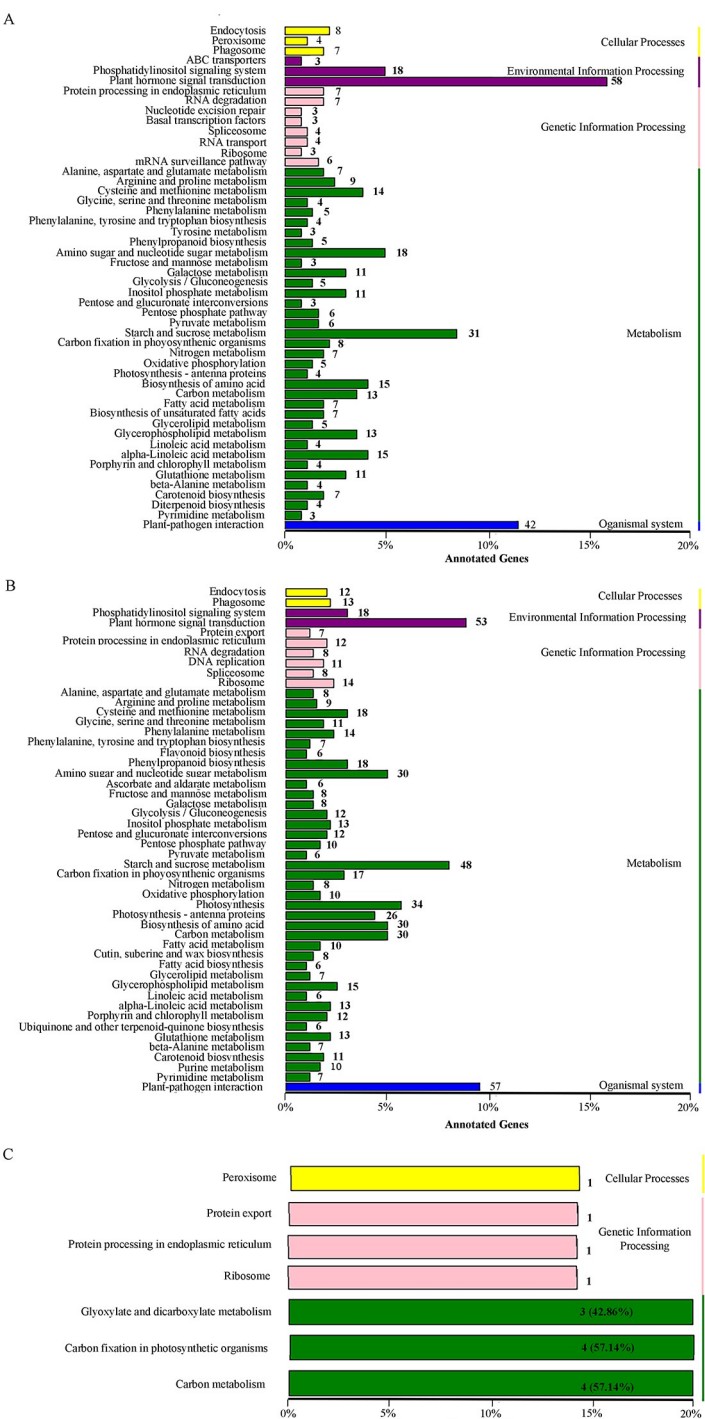

**Fig 4. KEGG classification map of the identified DEGs.** The ordinate shows the KEGG metabolic pathway, and the abscissa is the number of genes annotated to the pathway and the proportion of the number of genes annotated to the total number of genes annotated. A, KEGG classification of DEGs from the comparison SS-CK vs. SS-NT; B, KEGG classification of DEGs from the comparison ST-CK vs. ST-N; C, KEGG classification of DEGs from the comparison SS-NT vs. ST-NT. SS-CK and SS-NT indicate the salt-sensitive group samples treated without and with 150 mM NaCl; ST-CK and ST-NT indicate the salt-tolerant group samples treated without and with 150 mM NaCl.

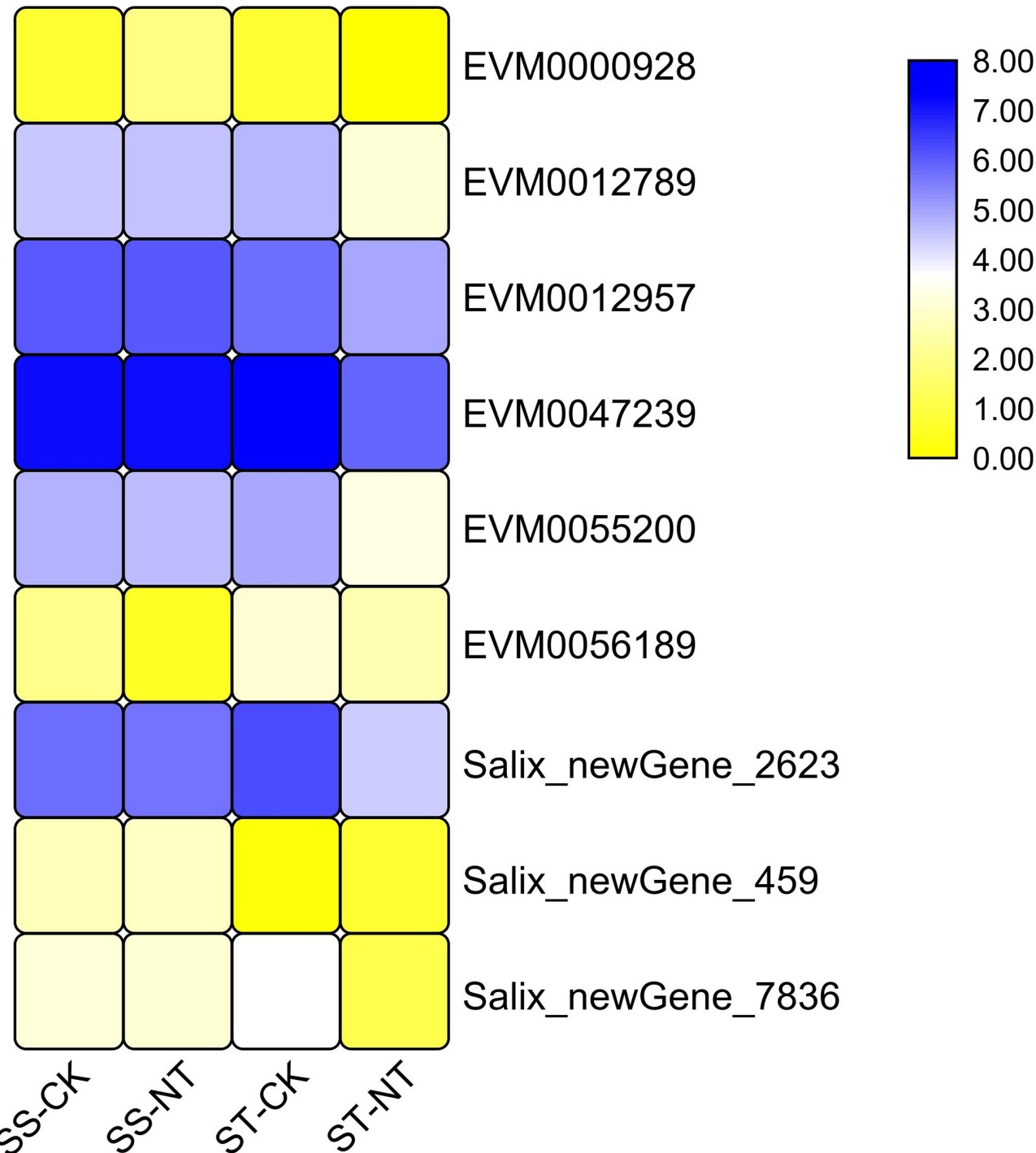

**Fig 5. Heatmap presentation of the DEGs from comparison SS-NT vs. ST-NT.** The average FPKM values of three repeat samples were counted. The heat map was drawn using Log10-transformed expression values. The color bar indicates the gene expression level. SS-CK and SS-NT indicate the salt-sensitive group samples treated without and with 150 mM NaCl; ST-CK and ST-NT indicate the salt-tolerant group samples treated without and with 150 mM NaCl. DEGs, differentially expressed genes.

apart from some housekeeping proteins such as the Calvin cycle protein, ribulose bisphosphate carboxylase small chain protein, and 60S ribosomal protein L12-like proteins, an AP2-like ethylene-responsive transcription factor and a sedoheptulose-1 family protein were also annotated, which were all previously reported as important players in mediating salt tolerance in *Arabidopsis* [37, 38].

## Several hub genes were identified by co-expression network analysis

A co-expression network through weighted gene co-expression network analysis (WGCNA) was constructed to uncover the interrelationships among salt responsive genes and determine the key regulators. Based on the co-expression relationships, several genes with the highest connectivity values in this network were identified as hub genes including EVM0006610, EVM0008812, EVM0009117, EVM0003949, EVM0005547, EVM0001761, EVM0006150, EVM0002493 and EVM0006603 (S2 Fig). EVM0006603 is a putative ethylene-responsive transcription factor.

## Identification of DEGs coding for TF with function in abiotic stress responses

Using HMM profile search against the database of *S. matsudana* protein, five TF families members related to salt stress tolerance were revealed. In total, 195 WRKYs, 365 R2-R3 MYBs, 167 bZIPs, 364 AP2/ERFs, and 292 NACs were found in the *S. matsudana* genome; the gene IDs are listed in S4 Table. A total of 38, 27, 15, 64, and 35 DEGs were identified in WRKY, R2R3-MYB, bZIPs, AP2/ERFs, and NAC TF families respectively; their expression level was up or downregulated after salt stress treatment (gene IDs and FPKM values are listed in S5 Table. The percentage of DEGs belonging to the WRKY TF family was 19.5%, while only 15 DEGs, accounting for 9%, were bZIP TFs. Among the four TF families of WRKY, bZIPs, AP2/ERFs, and NAC, the expression levels of most DEGs were increased after salt treatment, and only 4, 3, 8, and 8 were downregulated DEGs from these TF families respectively. Meanwhile, in R2R3-MYB TF family, 12 DEGs from a total of 27 DEGs were downregulated. As illustrated in Fig 6, the expression patterns of almost all TF DEGs after salt treatment were identical in the two samples of SS and ST, with only three exceptions found in WRKY (2 DEGs) and NAC (1 DEG) TF families. The expression of EVM0052749 and EVM0057278 from WRKY family was not induced in ST after salt treatment, but upregulated in '9901'. The expression of EVM0034132 was downregulated in SS after salt treatment but upregulated in ST (Fig 6).

## Identification of DEGs coding for important components of the salt stress response network

The *Arabidopsis* components/players from core stress signaling pathways of SOS, mitogen activated protein kinase cascades, and sucrose nonfermenting1-related protein kinase 2 (SnRK2)-mediated osmotic homeostasis, were firstly identified, and using them as queries, the homolog proteins from *S. matsudana* were revealed through BALSTP; the gene IDs and the FPKM values are listed in S6 Table. The pathway components, illustrated in Fig 7A, include SOS3, SOS2, SOS1, MAPKK cascade, ABA and its downstream SnRK2 pathway, $Na^+/H^+$ antiporters, antioxidase enzymes, and P5CS1, which catalyzes proline synthesis. The expression of PP2C (ABI2) in ABA signaling pathway, MAPKK cascade and several TFs including DREB1A, ESE1, ERF1 was increased after salt treatment. (Fig 7B). The salt treatments also enhanced the expression of $Na^+/H^+$ antiporters, including HTK and SOS1. Upon salt stress, $Ca^{2+}$ triggers the activation of respiratory burst oxidase homolog F/D (RbohF/D);

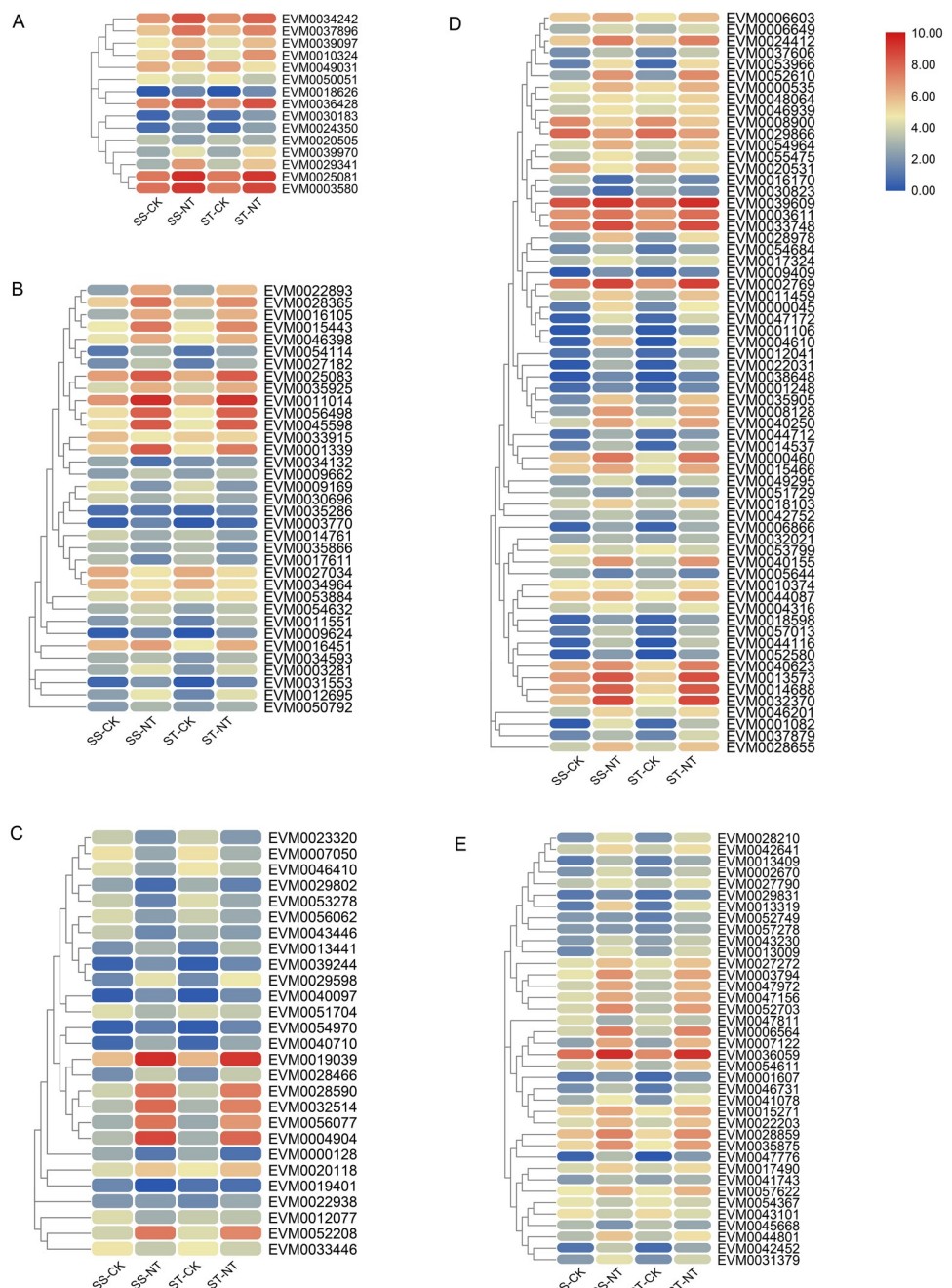

**Fig 6. Heatmap of the DEGs coding for five stress-response related TF family members.** Five stress-response related TF family members were screened from *Salix matsudana* genome; the TF DEGs were subjected to hierarchical clustering and heatmap presentation. The average FPKM values of three repeat samples were counted, and the heat map was drawn using Log10-transformed expression values. The color bar indicates the gene expression level. A, Heatmap for DEGs coding for bZIP; B, Heatmap for DEGs coding for NAC; C, Heatmap for DEGs coding for MYB; D, Heatmap for DEGs coding for AP2/ERF; E, Heatmap for DEGs coding for WRKY. DEGs, differentially expressed genes; TF, transcription factor. SS-CK and SS-NT indicate the salt-sensitive group samples treated without and with 150 mM NaCl; ST-CK and ST-NT indicate the salt-tolerant group samples treated without and with 150 mM NaCl. DEGs, differentially expressed genes.

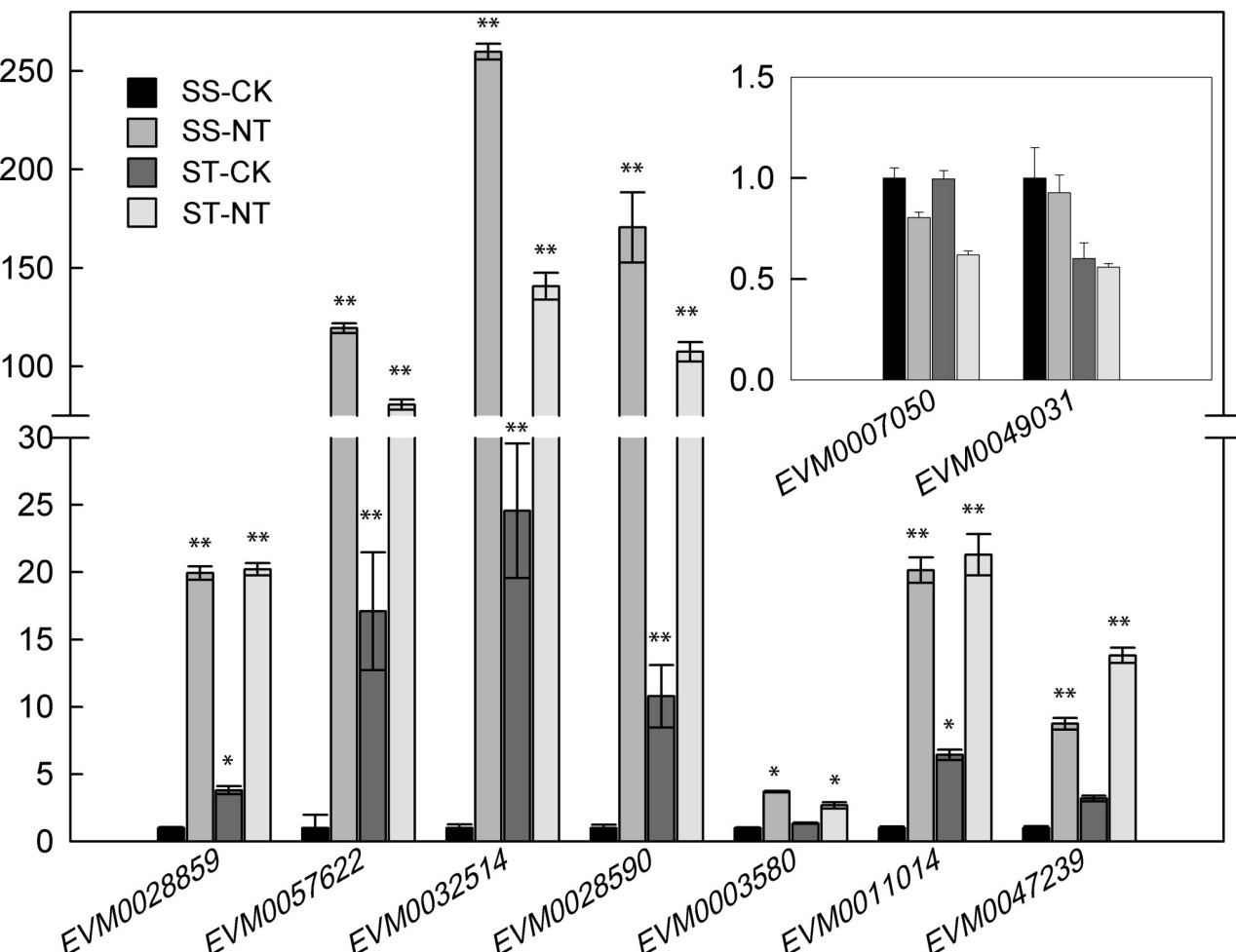

**Fig 7. DEGs coding for important components of the salt stress response network are illustrated in salt stress signaling pathway and differential expression patterns are presented by heatmap.** By homolog protein BLAST research, putative *Salix matsudana* stress signaling pathway players were revealed and showed in the salt stress network using cytoscape software, their expression levels were showed by heatmap. A, Schema of salt stress signaling network. The molecules labeled in purple indicate that their expression was upregulated after salt stress; those labeled blue indicate downregulation, and yellow indicates genes with no changes in expression level after salt stress. The shapes refer to different proteins categories. Octagons represent metabolism enzymes; ellipses represent kinases; arrowheads represent receptors; parallelograms represent transporters; round rectangles represent TFs and triangles represent other proteins including SOS3, CBL10, ABI2 and 14-3-3λ. B, Heat map showing the differential expression pattern of DEGs coding for proteins related to stress signaling pathway. The average FPKM values of three repeat samples were counted, and the heat map was drawn using Log10-transformed expression values. The color bar indicates the gene expression level. DEGs, differentially expressed genes. SS-CK and SS-NT indicate the salt-sensitive group samples treated without and with 150 mM NaCl; ST-CK and ST-NT indicate the salt-tolerant group samples treated without and with 150 mM NaCl. DEGs, differentially expressed genes.

this phenomenon was also reflected in our data. The mRNA level of the BAM enzymes, which degrade starch into sugar and sugar-derived, osmolytes, was induced by salinity stress (Fig 7B). The expression patterns of the ABA receptors, PYLs, were not identical; some were induced, and others were repressed or not change. No changes in the expression level of SOS3, SOS2, SnRK2 pathway upstream players, AREB/ABF, NHXs, antioxidase enzymes, and P5CS1 were detected after salt stress (Fig 7B). The expression patterns of the DEGs coding for molecular players in the salt resistance signaling pathway are demonstrated using a heatmap (Fig 7B).

## Validation of DEGs by qRT-PCR

To further validate the RNA-seq results, we conducted qRT-PCR analysis on nine DEGs; 7 were up-regulated and 2 were down-regulated DEGs. Furthermore, 8 of the DEGs coded for MYB, WRKY, NAC, or bZIP TFs, and 1 DEG coded for ribulose bisphosphate carboxylase small chain. The expression pattern of the genes in four willow samples were strongly correlated with the RNA-seq data, demonstrating the reliability of the RNA-seq results (Fig 8).

## Overexpression of *SBPase* gene in *Arabidopsis* enhanced the plant's salt tolerance ability

To determine the function of DEGs in plant salt stress responses, we selected the *SBPase* gene for further experimental analysis, because it was previously reported as an important player in oxidative stress and salt stress [34, 38]. *SBPase* gene was ectopically expressed in *Arabidopsis* under the control of the cauliflower mosaic virus 35S promoter. Using real-time PCR analysis, we found that the expression of *SBPase* gene in two overexpression lines was much higher than that in the control (Fig 9A). The SBPase activities were much higher in overexpression lines but deceased after salt treatments both in WT and transgenic lines (Fig 9B). After 24 h of salt treatment (200 mM), compared to the wild-type, the total soluble sugar content and sucrose content in the L20 and L22 transgenic lines were much higher, while the starch content in these transgenic lines was lower. After one week of salt stress, the total soluble sugar and sucrose contents were also increased in wild-type and the starch content was decreased, especially after 200 mM salt treatment (Fig 9C–9E). In wild type, the chlorophyll content was decreased after salt treatment but in overexpression lines, salt treatment enhanced the chlorophyll content (Fig 9F). The relative electrolyte leakage was increased to the similar level both in WT and overexpression lines after salt treatment (Fig 9G). Under salt stress, the seeds of L20 and L22 transgenic lines germinated earlier than WT (Fig 9H).

## Discussion

### Functional annotation and classification of assembled unigenes and DEGs

Salt tolerance signaling pathway is a complex network and includes a large number of genes. Understanding the mechanisms of salt stress response requires global information on stress-responsive genes. RNA-seq is a valuable tool that can uncover nearly the complete transcriptomic events after exposure to salt stress. Therefore, in this study, we characterized the salt stress response genes in *S. matsudana* using RNA sequencing. We identified 63409 putative unique transcripts, including 5568 new genes, in all sequencing samples and a total of 56624 genes were annotated by blasting against five databases. Notably, 56549 unigenes were annotated in the NR database, accounting for 99.8% of all the annotated genes. Many DEGs, including TFs and putative players in salt stress response network, were also identified, which will provide a basis for further studies. In this study, based on GO analysis of DEGs, several enriched biological processes were identified; for example, many DEGs were enriched in the GO term plant hormone signal transduction (GO: 0009737, response to abscisic acid; GO: 0009753, response to jasmonic acid). The enriched molecular function terms were related to DNA binding (GO: 0006355, regulation of transcription, DNA-templated), followed by catalytic activity. These GO annotations will provide insights and useful information for elucidating salt tolerance mechanisms and for finding new salt stress-related genes specific to *S. matsudana*.

KEGG enrichment analysis of the DEGs revealed several significantly enriched metabolic pathways and signal transduction pathways. These upregulated pathways include starch and

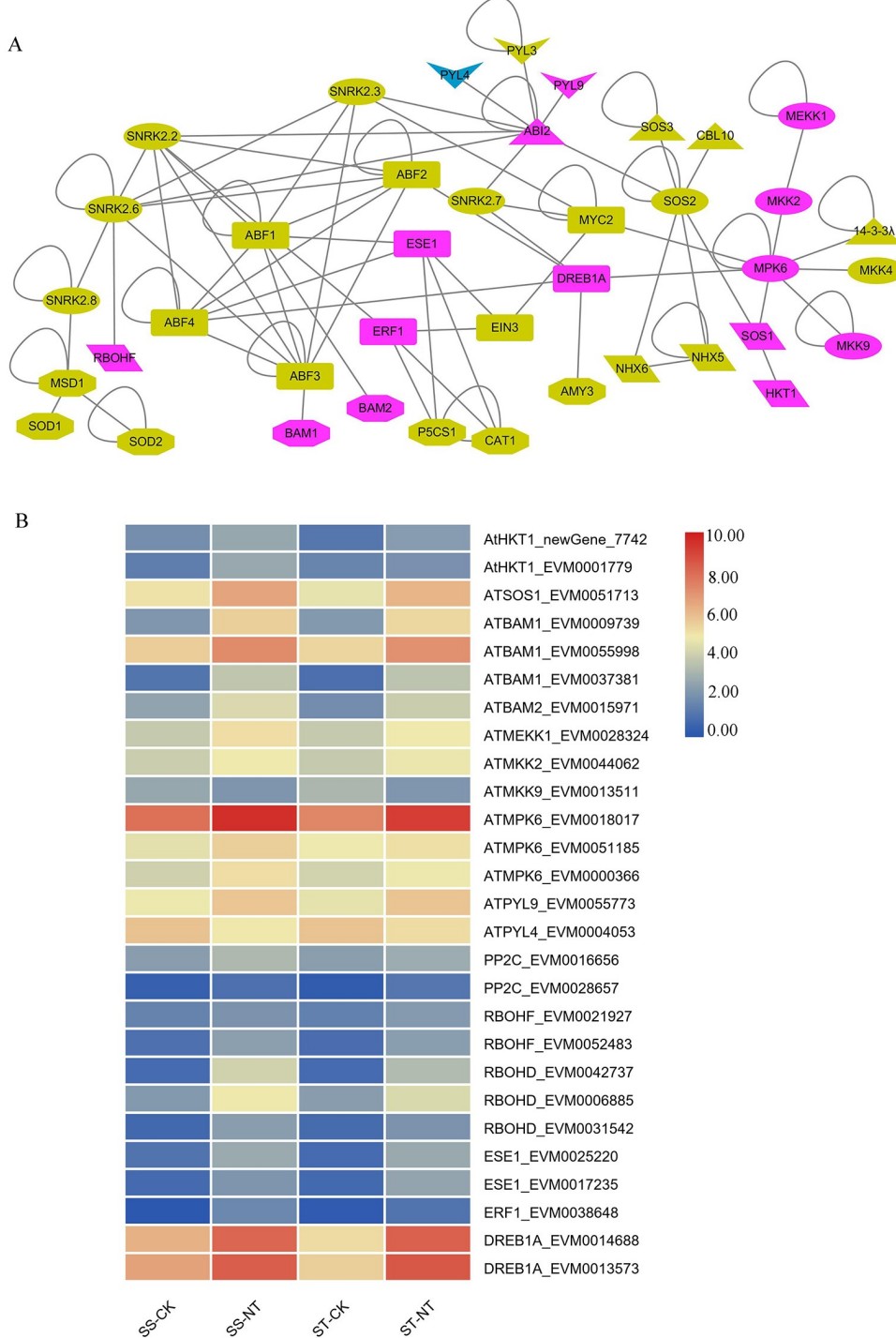

**Fig 8. Verification of 9 DEGs responsive to salt stress by quantitative real-time PCR (qRT-PCR).** For salt stress, 20-day hydroponic culture roots of 'SS' and 'ST' were treated with 150 mM NaCl for 4 h. The control was an untreated SS sample. Three biological replicates for each sample were performed and bars represent the standard deviations of the mean. ** $p < 0.01$ between treated sample and untreated control, Student's t-test. Gene expression profiles were evaluated using the $2^{-\Delta\Delta Ct}$ method and the value of control was normalized to 1. SS-CK and SS-NT indicate the salt-sensitive group samples treated without and with 150 mM NaCl; ST-CK and ST-NT indicate the salt-tolerant group samples treated without and with 150 mM NaCl. DEGs, differentially expressed genes.

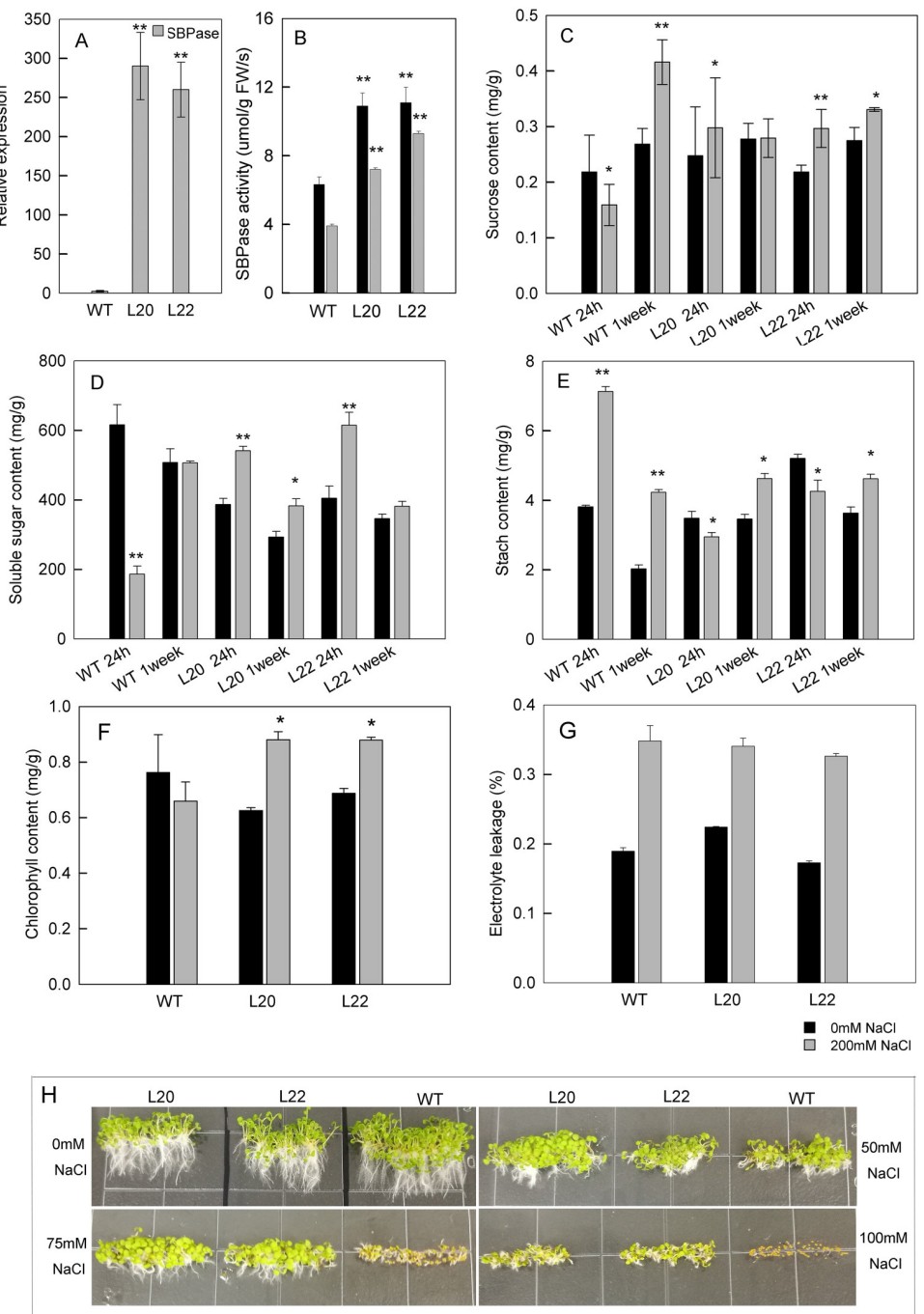

**Fig 9. Overexpression of *SBPase* gene in *Arabidopsis* enhances salt tolerance by increasing the sugar and chlorophyll content.** Two *Arabidopsis* transgenic lines (L20, L22) overexpressing the *SBPase* gene were generated by *Agrobacterium*-mediated transformation, and the SBPase activity, sucrose, soluble sugar, starch and chlorophyll content in these lines were determined. Relative electrolyte leakage was also detected. A, *SBPase* gene expression level in L20 and L22 transgenic lines. B, SBPase activities in L20 and L22 transgenic lines. C, Sucrose content of leaves from wild-type plants and two transgenic lines treated with salt stress for 24 h and 1 week; plants grown in normal condition were used as control. D, Soluble sugar content of leaves from wild-type and transgenic plants treated with salt stress for 24 h and 1 week; plants grown in normal condition were used as control. E, Starch content of leaves from wild-type and transgenic plants treated with salt stress for 24 h and 1 week; plants grown in normal condition were used as control. F, chlorophyll content of leaves from wild-type and transgenic lines plants treated with salt stress for 48 h. G, Relative electrolyte leakage of leaves from wild-type and transgenic lines plants treated with salt stress for 48 h. H, The seeds of WT and two transgenic Arabidopsis lines were planted on MS medium supplied with 0mM NaCl, 50mM NaCl, 75mM NaCl, 100mM NaCl respectively and cultured for 7 days.

sucrose metabolism, carbon fixation and photosynthetic metabolism, phosphatidylinositol signaling system, MAPK signaling pathway, calcium signaling pathway, and other secondary metabolite pathways, suggesting that maintaining osmotic balance and high photosynthetic metabolism efficiency play vital roles in salt stress tolerance in *S. matsudana*.

## Transcriptome analyses of TFs and their roles in the salt stress response of *S. matsudana*

TFs play indispensable roles in modulating the ionic, osmotic, and ROS balance in plants after salt stress. TFs involved in salt stress response mainly belong to five TF families, namely MYB, bZIP, NAC, AP2/ERF, and WRKY [39]. The family members from these five TFs were identified from *S. matsudana*. MYB family showed the largest representation among all five TF families, with 365 TFs in *S. matsudana* identified to belong to a subgroup of the MYB family, R2R3-MYB. A total of 195 WRKYs, 167 bZIPs, 364 AP2/ERFs, and 292 NACs were also identified in the *S. matsudana* genome. DEG analysis revealed that 19.5% of the WRKYs were DEGs, while only 9% of bZIPs were DEGs, indicating the important role of WRKYs in salt response. Most DEGs were upregulated in four TF families, while in the R2R3-MYB subfamilies, 12 DEGs from a total of 27 DEGs were downregulated (Fig 6). This special pattern might mean that the R2R3-MYB members may play complicated positive or negative roles in response to salt stress in *S. matsudana*. The expression patterns of almost all DEGs were similar in SS and NT, with only three exceptions: EVM0052749 and EVM0057278 from WRKY family and EVM0034132 from NAC family; whether these three genes confer the different salt tolerance capabilities of the two varieties need further experimental verification (Fig 6). A member of AP2/ERFs, EVM0006603 was identified as a hub gene in the co-expression (S2 Fig).

The discovery of the involvement of various TF families and different members of these families in the *S. matsudana* salt response indicates that a complicated transcriptional regulatory network is involved in this response. The identification of these TFs will be useful for studying the transcriptional regulatory switches involved in the adaptation of *S. matsudana* to environmental stress.

## Transcriptomic analyses of salt-responsive genes that encode important players of the salt stress response pathways

Several core stress signaling pathways participating in salt resistance have revealed by genetic and biochemical analyses in the past two decades [40]. The Salt Overly Sensitive signaling pathway plays a key role in maintaining ionic homeostasis, via extruding sodium ions into the apoplast. Mitogen activated protein kinase cascades mediate ionic, osmotic, and ROS homeostasis. SnRK2 (sucrose nonfermenting1-related protein kinase 2) proteins are involved in maintaining osmotic homeostasis. The major players in salt stress response identified in this study are listed in S6 Table and Fig 7. After receiving the salt stress signal(s), a series of signal transduction events occur sequentially, from $Ca^{2+}$ and ROS production, kinase activation, up or down regulation of TF genes and subsequent downstream gene expression, to maintaining the ionic, osmotic, and ROS homeostasis. In our transcriptomic analysis, DEGs were only found in some components of the signaling network. The first DEG group included five homolog genes of At RBOHF/D, which triggers the production of ROS; the expression of all five genes was increased after salt stress treatment. The second DEG group included genes coding for MAP cascade kinases, which are key players in the salt stress signaling pathway. The expression levels of all DEGs in this group were enhanced by salt stress. The third group of DEGs was the $Na^+/H^+$ antiporters including HTK and SOS1. All three detected DEGs in

this group were upregulated. The fourth upregulated DEG group included genes coding for BAMs, which degrade starch into sugar and sugar-derived osmolytes to maintain the osmatic balance. PYLs and PP2Cs, components of the ABA signaling pathway, which is involved in salt stress response, were also detected as DEGs. To alleviate the damage of salt stress to cells, the expression of SOS1 needs to be upregulated to export more $Na^+$ to apoplast; on the contrary, the expression of HTK1 should be repressed to inhibit the transport of $Na^+$ from xylem to cell. As illustrated in Fig 7, the expression of SOS1 was upregulated but the expression of HTK1 was not repressed at 4 h after salt stress treatment, just as the expression of P5CS1 and antioxygen enzymes genes, which might be the downstream genes in the salt stress signaling pathway, were not increased at the time detected.

### '9901' was more tolerant to salt stress than 'Yanjiang' because of a set of DEGs

Although, based on the phenotype analysis, we could infer that under salt stress, '9901' presents higher slat tolerance capability than 'Yanjiang', only some DEGs were obtained between the salt-sensitive and the salt-tolerant samples under salinity stress (Fig 5). This may be due to the short salt treatment time or *S. matsudana* itself has certain salt tolerance ability. From the transcriptome analysis, several DEGs were identified, which might provide the clues for the higher salinity tolerance of '9901'. Firstly, from the 8 downregulated DEGs revealed from pair-wise combination SS-NT vs ST-NT, 5 genes were found coding for carbon metabolism pathway enzymes, which indicated that the plant acclimation (reduced growth) after salt stress depended ultimately on alterations in photosynthetic metabolism (S3 Table) [39]. Secondly, from the salt signaling pathway, expression levels of HKT1 and AtRBOHF/D homolog genes were lower in ST-NT, which might result in the reduced import of $Na^+$ into cell and also produce less ROS in cell [34]. These mechanisms all alleviate the damage of salt stress to cell.

The differential expression patterns of EVM0052749 and EVM0057278 from WRKY family and EVM0034132 from NAC family in SS and ST might be another reason for the different salt tolerance capabilities of the two willow varieties. Except for nine genes identified in SS-NT vs ST-NT, more attentions should be made on other DEGs because more DEGs were found in ST-CK vs ST-NT comparison than that found in SS-CK vs SS-NT, DEGs from five different TF families and key genes in co-expression network may also account for the stronger salt tolerance of '9901'.

### Functional analysis of *SBPase* gene revealed that RNA-Seq is a dependable strategy to uncover important genes in salt tolerance

The *SBPase* gene was a DEG selected based on pair-wise combination of SS-NT vs ST-NT. Ectopic overexpression of this DEG in *Arabidopsis* conferred higher salt tolerance to the *Arabidopsis* transgenic lines. Treatment with high salt concentration solution rapidly triggered salt stress response in the transgenic lines by enhancing chlorophyll content and degrading starch to promote the accumulation of soluble sugars, including sucrose. Soluble sugars are osmolytes that can maintain the osmatic balance, thereby enhancing the salt tolerance ability. Higher chlorophyll content in transgenic plants means higher photosynthesis capability after salt stress. Seed germination experiment further showed that two transgenic lines have higher salt tolerance capability than WT. Our functional analysis on the *SBPase* gene also suggested that RNA-seq is a dependable strategy to uncover important genes in salt tolerance.

## Conclusions

Through RNA sequencing, DEGs were identified from comparison between *S. matsudana* salt-tolerant and salt-sensitive samples. The DEGs were enriched in several pathways, including carbon metabolism pathway, plant-pathogen interaction pathway, and plant hormone signal transduction pathway. DEGs coding for TFs with functions in abiotic stress responses and important components of the salt stress response network were identified and their expression levels differed between the two samples in response to salt stress. Functional analysis on *SBPase* gene via transgenic expression of the gene in *Arabidopsis* showed that increased SBPase activity could increase the photosynthetic rates and sucrose and starch accumulation, leading to enhanced salt tolerance.

## Supporting information

**S1 Fig. Cluster map of differentially expressed genes from three comparisons.**
(TIF)

**S2 Fig. Co-expression network of DEGs revealed by weighted gene co-expression network (WGCNA) analysis.**
(PDF)

**S1 Table. Primer list for qRT-PCR analysis.**
(DOCX)

**S2 Table. FPKM values of all genes from RNA sequencing.**
(XLS)

**S3 Table. The GO, KEGG pathway, and NR annotation of differentially expressed genes in salt-sensitive and salt-tolerant comparison groups under 150 mM NaCl treatment.**
(DOCX)

**S4 Table. The list of genes related to five transcription factor families, including MYB, AP2ERF, NAC, WRKY and bZIP in *S. matsudana*.**
(XLSX)

**S5 Table. Differentially expressed genes belonging to five transcription factor families, including MYB, AP2ERF, NAC, WRKY and bZIP in *S. matsudana*.**
(XLSX)

**S6 Table. List of genes associated with salt signaling pathways in *S. matsudana* and list of differentially expressed genes.**
(XLSX)

## Acknowledgments

We thank editing service from Editage Company (https://www.editage.cn) for professional scientific editing the English text of a draft of this manuscript.

## Author Contributions

**Conceptualization:** Yanhong Chen, Jian Zhang.

**Data curation:** Yanhong Chen, Yu Chen, Guoyuan Liu, Chunmei Yu.

**Investigation:** Yuna Jiang, Yu Chen.

**Methodology:** Yuna Jiang, Yu Chen, Wenxiang Feng, Guoyuan Liu, Chunmei Yu, Bolin Lian, Fei Zhong.

**Software:** Guoyuan Liu.

**Validation:** Wenxiang Feng.

**Writing – original draft:** Yanhong Chen, Yu Chen.

**Writing – review & editing:** Yanhong Chen, Jian Zhang.

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
