## [Decision Letter · Decision Letter 0]

6 May 2020

PONE-D-20-09667

Uncovering candidate genes responsive to salt stress in Salix matsudana (Koidz) by transcriptomic analysis

PLOS ONE

Dear Dr Zhang,

Thank you for submitting your manuscript to PLOS ONE. After careful consideration, we feel that it has merit but does not fully meet PLOS ONE’s publication criteria as it currently stands. Therefore, we invite you to submit a revised version of the manuscript that addresses the points raised during the review process.

To enhance the reproducibility of your results, we recommend that if applicable you deposit your laboratory protocols in protocols.io, where a protocol can be assigned its own identifier (DOI) such that it can be cited independently in the future. For instructions see: http://journals.plos.org/plosone/s/submission-guidelines#loc-laboratory-protocols

We look forward to receiving your revised manuscript.

Kind regards,

Jin-Song Zhang, Ph.D.

Academic Editor

PLOS ONE

Journal Requirements:

1. We noticed you have some minor occurrence of overlapping text with the following previous publication(s), which needs to be addressed:

- https://www.researchsquare.com/article/rs-15225/v1

- http://www.funpecrp.com.br/gmr/year2016/vol15-3/pdf/gmr8738.pdf

- https://journals.plos.org/plosone/article/authors?id=10.1371/journal.pone.0171451

In your revision ensure you cite all your sources (including your own works), and quote or rephrase any duplicated text outside the methods section. Further consideration is dependent on these concerns being addressed.

3. Please amend the manuscript submission data (via Edit Submission) to include author Guoyuan Liu

4. Please amend your authorship list in your manuscript file to include author Guoliang Liu

Reviewers' comments:

Reviewer's Responses to Questions

**Comments to the Author**

1. Is the manuscript technically sound, and do the data support the conclusions?

Reviewer #1: Yes

Reviewer #2: Yes

2. Has the statistical analysis been performed appropriately and rigorously? 

Reviewer #1: Yes

Reviewer #2: Yes

3. Have the authors made all data underlying the findings in their manuscript fully available?

Reviewer #1: Yes

Reviewer #2: Yes

4. Is the manuscript presented in an intelligible fashion and written in standard English?

Reviewer #1: Yes

Reviewer #2: No

5. Review Comments to the Author

Reviewer #1: 1, The author mentioned the use of Nikon Z50 instrument for analysis root phenotype, but did not provide specific manufacturers and country, please add details.

2, The author mentioned that “These clean reads were then mapped to the reference genome sequence.” Which species conference genome is the so-called reference genome? Willow or poplar?

3, Why only select SBP gene for functional analysis by transgenic Arabidopsis? We can use yeast salt sensitive mutants for expression analysis, and then select key genes for Arabidopsis functional verification. And the phenotype of transgenic Arabidopsis was not provided before and after salt stress.

4, What is the role of SBPase gene in Fig8A? I can’t seem to be reflected in this picture? This gene is a osmo-regulation gene, and this pathway is not found in the Fig 8A.

5, In the MS, the data mining of transcriptome is not enough. The co-expression network can be constructed using transcriptome data and then what the hub genes are selected. This result is more convincing for the key genes selected.

Reviewer #2: In the manuscript “Uncovering candidate genes responsive to salt stress in Salix matsudana (Koidz) by transcriptomic analysis”, the authors compare the transcriptomes of two Salix matsudana cultivars, i.e., “9901” and “Yanjiang”, thus identifying salt stress responsive genes and genes determining the different salt tolerance in Salix matsudana. The following qRT-PCR of DEGs validates the accuracy of RNA-seq data. Overexpression of the DEG SBPase enhances the salt tolerance in transgenic plants, further suggesting that the identification of candidate genes is highly efficient and credible. This study contributes to dissect the molecular mechanism of salt tolerance on Salix matsudana. The critical genes involved in salt tolerance may be useful for willow breeding.

I have couples of doubts and comments:

The manuscript has serious problems with English language, and needs to be revised and rewritten from this perspective. Also, there are lots of inconsistencies and errors in the context. For instance, the name of salt-sensitive material is not consistent in the manuscript. Is it “Yanjiang”, “Yanjang” or “Yanjing”? Second, figure 7 and 8 should switch the figure legend. Moreover, the authors ignore to number the line that makes it difficult to label the errors in the manuscript.

The authors mention 15 individual lines including 12 F1 lines from crossing of “9901” and “Yanjiang” and 3 parent lines in plant material, but only “9901” and “Yanjiang” are used as described in result.

The difference on shooting pattern should be taken into account for the determination of salt tolerance between “9901” and “Yanjiang”. The stems grow in absence of NaCl are necessary for the control in this case. Since root regeneration of Salix matsudana is possible, rooted plants would be better for testing phenotype as root material is also used for RNA-seq.

What samples are used in qRT-PCR? Please clarify.

It is good that homologs of known and important salt responsive regulators in Salix matsudana are identified from RNA-seq data. however, to better illustrate their interactions under salt stress, I would also suggest to visualize the network by Cytoscape as described by Li et al., 2013 other than the schematic diagram and heatmap. The potential interactors of those components might also be mined by this manner.

Sugar content is not enough to determine the salt tolerance of transgenic Arabidopsis plant although SBPase is known to be involved in carbon assimilation. Instead, the survival rate, electrolyte leakage and chlorophyll content under salt stress should be tested. In addition, please specify how SBPase is regulated by salt stress here.

The second paragraph of discussion is basically repeating the result, lacking the further prediction of functions of TFs discovered in the RNA-seq. Jia et al. studied drought response of a desert willow which revealed important roles of TFs in the regulatory network. The authors could further compare differentially expressed TFs under salt and drought stress within willow species, thereby illustrating the identical and different strategies how willow fights against various abiotic stresses.

Except for nine genes identified in “Yanjiang” NT vs “9901” NT, the other DEGs in “9901” CK vs “9901” NT may also account for the stronger salt tolerance of “9901” as “9901” has more DEGs compared with “Yanjiang” in response to salinity. Detailed discussion is necessary on this point.

Minor points:

Please cite the literature where the agrobacteria-mediated protocol comes from.

What does “NT” indicate in the legends of Figure 3 and Figure 4?

Do asterisks in figure 9 indicate significant differences?

Reference

Li, Q.T., Liu, J., Tan, D.X., Andrew, A.C., Jiang, Y.Z., Xu, X.F., Han, Z.H. and Kong, J. (2013) A genome-wide expression profile of salt-responsive genes in the apple rootstock Malus zumi. Int. J. Mol. Sci. 14, 21053-2107.

Jia, H., Zhang, J., Li, J., Sun, P., Zhang, Y., Xin, X., Lu, M., and Hu, J. (2019). Genome-wide transcriptomic analysis of a desert willow, Salix psammophila, reveals the function of hub genes SpMDP1 and SpWRKY33 in drought tolerance. BMC Plant Biol. 19, 356.

6. PLOS authors have the option to publish the peer review history of their article (what does this mean?). If published, this will include your full peer review and any attached files.

Reviewer #1: No

Reviewer #2: Yes: Qingtian Li

---

## [Author Response · Author response to Decision Letter 0]

9 Jun 2020

Response to Reviewer's 1

1. We provided specific manufacturers and country to the Nikon Z50 instrument according to review’s suggestion.

2. The conference genome we mentioned in the manuscript is genome Salix matsudana which sequenced recently in our lab but not published yet.

3. The SBPase is an enzyme coded by a DEG selected from pairwise combination salt-treated salt-sensitive vs salt-treated salt-tolerant. It was previously reported that this gene was important players in mediating salt tolerance in Arabidopsis, so we selected this gene to do further functional analysis. In many previous reports, researchers usually use Arabidopsis or tobacco to do ectopic functional verification on some genes from species which can’t obtain transgenic plants. There are no proper yeast strains to test the gene’s function in our lab, so we directly using the Arabidopsis. In the resubmission manuscript, we add the data concerning the phenotype of transgenic Arabidopsis plant, such as the survival rate and chlorophyll content under salt stress.

4. As what the reviewer’s said, the role of SBPase gene is concerning the carbohydrate metabolism and osmotic –regulation. The Fig 8A demonstrated the important players in salt stress signaling pathway and corresponded the result part “Identification of DEGs coding for important components of the salt stress response network”. So we didn’t include the SBPase gene in the Fig 8A.

5. According to the reviewer’s comments, using transcriptome data, the Weighted gene co-expression network (WGCNA) analysis was performed and several hub genes were revealed in the new submission （S2 Figure）, which will guide our future work in gene functional analysis. 

Response to Reviewer's 2

1. We edited the language usage, spelling, and grammar in my manuscript and corrected the inconsistencies and errors reviewer mentioned. “Yanjiang” is the correct name for the salt-sensitive Salix matsudana material.

2. We realized that using “9901” and “Yanjiang” in the result is not reasonable for the salt-tolerant and salt-sensitive material, we have edited the description in the resubmission manuscript, Salt sensitive (SS) refers to Yanjiang and salt-sensitive lines from F1; Salt tolerant (SN) refers to 9901 and salt-tolerant lines from F1.

3. In the figure 1, the stems of salt-tolerant and salt-sensitive lines were directly immersed into the 100mMol NaCl, it’s difficult for the stems to root in this concentration. But 9901 can shoot normally, which showed the difference capability in salt tolerance. Without salt stress, all stems can root normally. In the RNA sequencing, the experiments were carried out as the description in Materials and methods, after culture in solution without NaCl for 20d, the roots were generated and the stems with roots were transferred to 150 mM NaCl to do salt stress treatment.

4. The root samples were used in qRT-PCR. The root samples came from the repeated experiments which mimic the RNA sequencing material collection procedure. 

5. According to the reviewer’s comments, we added the network diagram created by Cytoscape in the new submission and the reference was also added in the reference list.

6. According to the reviewer’s comments, more analysis on the survival rate, electrolyte leakage, SBPase activity and chlorophyll content under salt stress of transgenic Arabidopsis plant were made. In our experiment, the 3 weeks seedlings of WT and overexpression lines were watering with 200mM NaCl for 2 Weeks, but no evident difference in survival rate was found, almost all seedlings are alive (Perhaps because of the short treatment time or low NaCl concentration). Relative electrolyte leakage was increased both in WT and overexpression lines, and no difference was not found either. The only difference was found in chlorophyll content; the content in overexpression lines was higher than that in WT lines after salt treatment. 

7. Only depending on the expression pattern, it’s hard to make deep prediction of functions of TFs discovered in the RNA-seq, but we rewrite this part in the new submission to make some comparison and detailed discussion. Uncovering the important roles of TFs under salt and drought stress within willow species is our aims of future research. 

8. We agree with the opinion of reviewer that “other DEGs in “9901” CK vs “9901” NT may also account for the stronger salt tolerance”. In the manuscript, we made some discussion on this part DEGs including five different TF family members and player’s in salt stress signaling pathway.

---

## [Decision Letter · Decision Letter 1]

16 Jun 2020

PONE-D-20-09667R1

Uncovering candidate genes responsive to salt stress in Salix matsudana (Koidz) by transcriptomic analysis

PLOS ONE

Dear Dr. Zhang,

Thank you for submitting your manuscript to PLOS ONE. After careful consideration, the reviewer 2 is still not satisfied. I agree with his comments. Please provide a photo for both materials used and show the regeneration of shoots under normal growth condition. This photo should be placed in Fig1. In addition, please provide pictures of the two transgenic Arabidopsis lines in comparison to the WT to show salt tolerance under salt stress condition. The growth of these plants under normal growth condition should also be provided for comparison. These may be placed in Fig9. Moreover, please recheck the English writing as many mistakes were noted. Specifically, line 486, the 'oxidase stress' should be 'oxidative stress'. Line 503, 'was' should be 'were'. Line 505, the second 'were' should be 'was'. Please check all the other text, legends and method section etc. 

We look forward to receiving your revised manuscript.

Kind regards,

Jin-Song Zhang, Ph.D.

Academic Editor

PLOS ONE

Reviewers' comments:

Reviewer's Responses to Questions

**Comments to the Author**

1. If the authors have adequately addressed your comments raised in a previous round of review and you feel that this manuscript is now acceptable for publication, you may indicate that here to bypass the “Comments to the Author” section, enter your conflict of interest statement in the “Confidential to Editor” section, and submit your "Accept" recommendation.

Reviewer #1: All comments have been addressed

Reviewer #2: (No Response)

2. Is the manuscript technically sound, and do the data support the conclusions?

Reviewer #1: Yes

Reviewer #2: Yes

3. Has the statistical analysis been performed appropriately and rigorously? 

Reviewer #1: Yes

Reviewer #2: Yes

4. Have the authors made all data underlying the findings in their manuscript fully available?

Reviewer #1: Yes

Reviewer #2: Yes

5. Is the manuscript presented in an intelligible fashion and written in standard English?

Reviewer #1: No

Reviewer #2: Yes

6. Review Comments to the Author

Reviewer #1: Based on the analysis of transcriptome of willow under salt stress, the molecular bsis of salt tolerance was identified, and the preliminary functional verification of the SBP gene was carried out. The author had made the changes according to the reviewer's comments

Reviewer #2: In the revised manuscript, the authors present a clearer transcriptomic analysis on salt-responsive mechanism of Salix matsudana compared with the original version, however, fail to respond to my minor points.

I believe 9901 and Yanjiang shoot normally without NaCl but the images of control are necessary for the manuscript.

As 200 mM NaCl does not make an obvious difference on survival rate between WT and SBPase transgenic Arabidopsis plant, I do not think any more experiments are necessary on this point, but I would encourage the authors to try a higher concentration of NaCl because it suggests that SBPase might not be that important for salt-tolerance if transgenic plant does not show higher survival rate in comparison with WT. Alternatively, the authors could measure the biomass of plants under 200 mM NaCl treatment. Also, please specify the background of Arabidopsis used for transformation.

7. PLOS authors have the option to publish the peer review history of their article (what does this mean?). If published, this will include your full peer review and any attached files.

Reviewer #1: Yes: Renying Zhuo

Reviewer #2: Yes: Qingtian Li

---

## [Author Response · Author response to Decision Letter 1]

26 Jun 2020

Response to Reviewer's 2

1. In the new submission, the images of ‘9901’ and ‘Yanjiang’ materials which cultured under normal growth condition were used as control and placed in Fig1 as Fig1 E and Fig1 F respectively.

2. In the new submission, we changed the experiment strategy by planting the seeds directly on MS medium supplied with 0mM NaCl, 50mM NaCl, 75mM NaCl, 100mM NaCl respectively and cultured for 7 days. The phenotype differences were recorded and the pictures were placed in Fig9 as Fig9 H. Results showed that compared with WT plants, the two transgenic Arabidopsis lines had high salt tolerance capability with earlier germination and higher growth rate in medium with higher salt content.

3. In the manuscript, Arabidopsis ecotype Col-0 was used to do transformation experiments.

---

## [Editor Report · Decision Letter 2]

30 Jun 2020

Uncovering candidate genes responsive to salt stress in Salix matsudana (Koidz) by transcriptomic analysis

PONE-D-20-09667R2

Dear Dr. Zhang,

We’re pleased to inform you that your manuscript has been judged scientifically suitable for publication and will be formally accepted for publication once it meets all outstanding technical requirements.

Kind regards,

Jin-Song Zhang, Ph.D.

Academic Editor

PLOS ONE
---

## [Editor Report · Acceptance letter]

6 Jul 2020

PONE-D-20-09667R2 

Uncovering candidate genes responsive to salt stress in Salix matsudana (Koidz) by transcriptomic analysis 

Dear Dr. Zhang:

I'm pleased to inform you that your manuscript has been deemed suitable for publication in PLOS ONE. Congratulations! Your manuscript is now with our production department. 

Kind regards, 

on behalf of

Prof. Jin-Song Zhang 

Academic Editor

PLOS ONE